# Approaches to Manipulate Ephrin-A:EphA Forward Signaling Pathway

**DOI:** 10.3390/ph13070140

**Published:** 2020-06-30

**Authors:** Sarah Baudet, Johann Bécret, Xavier Nicol

**Affiliations:** Institut de la Vision, Sorbonne Université, Inserm, CNRS, 17 rue Moreau, F-75012 Paris, France; sarah.baudet@inserm.fr (S.B.); johann.becret@inserm.fr (J.B.)

**Keywords:** ephrin, Eph, second messengers, peptides, vav guanine nucleotide exchange factor, Ephexin, cyclic guanosine monophosphate, cyclic adenosine monophosphate, calcium

## Abstract

Erythropoietin-producing hepatocellular carcinoma A (EphA) receptors and their ephrin-A ligands are key players of developmental events shaping the mature organism. Their expression is mostly restricted to stem cell niches in adults but is reactivated in pathological conditions including lesions in the heart, lung, or nervous system. They are also often misregulated in tumors. A wide range of molecular tools enabling the manipulation of the ephrin-A:EphA system are available, ranging from small molecules to peptides and genetically-encoded strategies. Their mechanism is either direct, targeting EphA receptors, or indirect through the modification of intracellular downstream pathways. Approaches enabling manipulation of ephrin-A:EphA forward signaling for the dissection of its signaling cascade, the investigation of its physiological roles or the development of therapeutic strategies are summarized here.

## 1. Introduction

Eph (erythropoietin-producing hepatocellular carcinoma) receptors belong to a family of receptor tyrosine kinases (RTKs) divided in two distinct classes (A and B). EphAs mostly bind ephrin-As ligands, a family of 5 glycosylphosphatidylinositol (GPI)-anchored proteins, except for EphA4, which binds ephrin-B2 and B3 ligands. The specificity of each EphA for individual ephrin-As is low, enabling a redundancy between the functions of individual ephrin-As and EphAs. Only EphA1 binds exclusively one ephrin-A: ephrin-A1. Ephrin-As have initially been described as ligands of their EphA receptors, initiating a signaling cascade in the EphA-carrying cell (forward signaling). However, ephrin-As can also act as a receptor, transducing a signal when bound to EphAs acting as ligands (reverse signaling) [1]. This review focuses on EphA forward signaling and on available approaches to manipulate it.

## 2. Physiology of Ephrin-A:EphA Forward Signaling

### 2.1. Development of the Nervous System

Ephrin-A:EphA signaling has been widely studied in the developing nervous system, at embryonic and early post-natal stages. Ephrin-As and EphAs are highly expressed during these developmental stages. They are involved in early events shaping the developing embryo. EphA4 is critical for cell sorting and the formation of sharp tissue boundaries, often in combination with its ephrin-B2 and B3 ligands. These interactions rely on complementary expression patterns between the ephrin ligand and the Eph receptor. For instance, the development of the ectoderm-mesoderm boundary and the segmentation of the hindbrain require EphA4 signaling [2,3,4]. In the developing nervous system, ephrin-As and EphAs are involved in neurogenesis, neural migration, axon guidance, exuberant connection pruning, and synaptogenesis [5,6]. In the developing cerebral cortex, excitatory neurons migrate radially from a proliferative zone located in the vicinity of the cerebral ventricle and form functional columns assembling mostly neuron from the same proliferative site. However, a few neurons migrate laterally and reach neighboring columns. This lateral dispersion contributes to the interconnection of cortical columns and requires ephrin-A:EphA forward signaling [7]. In contrast to excitatory neurons, cortical inhibitory neurons originate from the ganglionic eminence in the ventral telencephalon and undergo tangential migration to reach their integration site. Ephrin-A:EphA forward signaling contributes to keeping migrating interneurons on their path towards the cortex [8,9]. Later during development, ephrin-As repel EphA-expressing axons, orienting their growth and indicating the position of their terminal arbor. Ephrin-A-induced axon repulsion has been extensively studied in the context of retinal topographic map development. A gradient expression of EphAs in retinal axons together with the expression pattern of ephrin-A ligands in their specific brain targets namely, the superior colliculus and the dorso-lateral geniculate nucleus, dictate the position of retinal axon termination zones in these regions [10]. This mechanism has been extended to many other projection neurons including spinal motor neurons [11,12]. Once axons have reached their targets, synapses are formed with their post-synaptic partners. The maturation of synapses in the developing cerebral cortex is influenced by EphA7 signaling [13]. Apoptosis is a critical process to modulate the number of neurons and progenitor survival in the developing nervous system. EphA7 and ephrin-A5 modulate caspase pathways and apoptosis in a diversity of regions of the developing brain [14,15,16]. The ephrin-A5:EphA7 pathway induces the death of retinal and cortical progenitors in a caspase-3-dependent manner during development. Stimulating this pathway leads to a reduction of the size of the retinal and the cerebral cortex via the reduction of the progenitor pool [16,17]. In contrast, knocking-out EphA7 prevents the cell death of the progenitors and induces the enlargement of the cortex [17].

### 2.2. Adult Nervous System

Although EphA and ephrin-A expression is largely downregulated in the adult nervous system, it is maintained in regions where developmental events are still present, including plastic synapses and neural stem cell niches. Synaptic plasticity, although reduced compared to developmental stages, is still present in the adult nervous system. Ephrin-A3 expressed by glial cells in the hippocampus signals through neuronal EphA4 to control synaptic morphology and plasticity [18,19]. Similarly, glial ephrin-A2 reduces synaptic pruning in the adult cerebral cortex [20]. In neural stem cell niches, while EphA4 forward signaling is involved in the maintenance of stem cell fate [21], ephrin-A2 and A3 expression both in neurogenic and non-neurogenic regions reduces the proliferation of neural progenitor cells in adults [22,23].

### 2.3. Outside the Nervous System

The impact of ephrin-A:EphA signaling on normal development and physiology is not restricted to the nervous system. Among others, angiogenesis, insulin secretion, morphogenesis in a diversity of tissue, and bone homeostasis are influenced by ephrin-A forward signaling. EphA2 is expressed in the angiogenic vasculature and enhances neovascularization upon ephrin-A binding [24,25,26]. EphA signaling enhances the VEGF-induced neovascularization in the retina and the cornea [25,26]. This EphA pathway involves the guanine nucleotide exchange factors Vav2 and Vav3 [24]. Glucose-induced Ephrin-A5 forward signaling is a negative regulator of insulin secretion by pancreatic islets. Interestingly, pancreatic β cells use EphA forward signaling to inhibit insulin secretion and downregulate this pathway while shifting to ephrin-A reverse signaling when insulin is required for glucose homeostasis [27]. EphA kinase activation regulates epithelial branching in a diversity of epithelia including the mammary gland and the kidney where it negatively regulates hepatocyte growth factor (HGF)-induced branching [28,29]. EphA2-deficient animals exhibit a reduced mammary epithelium growth and branching into the fat pad [28], and ephrin-A1 reduces the branching of epithelial kidney cells induced by HGF [29]. Bone homeostasis is regulated by a balance between osteoclast and osteoblast activity. While osteoblasts promote bone formation, osteoclasts are responsible for bone resorption. Ephrin-A2:EphA2 signaling in osteoclasts and osteoblasts negatively regulates bone formation by enhancing osteoclastogenesis and reducing osteoblastogenesis [30,31].

### 2.4. Ephrin-A:EphA Signaling in Pathological Conditions

Although mostly expressed during development, the ephrin-A:EphA system is reactivated in adults in pathological conditions. Traumatic or ischemic lesions of the central nervous system often lead to upregulation of EphAs and ephrin-As, contributing to the scarce regeneration of injured axons. Preventing expression of EphA4 or ephrin-A3 enhances axon growth after optic nerve lesion [32] and EphA4 blockade promotes axon regeneration in the injured spinal cord [33,34]. Deletion of the EphA2 gene reduces the injury-induced by ischemia in the brain [35]. Several members of the ephrin-A system including EphA1, EphA4, ephrin-A1, and ephrin-A5 have been associated with a diversity of neurodegenerative conditions such as Alzheimer’s disease or amyotrophic lateral sclerosis. For instance, EphA4 is a substrate of γ-secretase, a protease dysfunctioning in many early-onset Alzheimer’s disease cases [36], and this member of the EphA family controls the metabolism of the amyloid precursor protein [37]. Reduced EphA4 expression is associated with enhanced survival of amyotrophic lateral sclerosis patients and in animal models of this pathology [38].

Outside the nervous system, ephrin-A:EphA signaling is implicated in a diversity of pathological conditions. In the heart, ephrin-A forward signaling promotes the regenerative ability of cardiac progenitor cells [39]. Acute lung injury involves EphA2 that antagonizes inflammation, pulmonary vascular permeability, and oxidative stress [40]. Eph receptors have been first identified in the context of cancer in the erythropoietin-producing hepatoma cell line and the impact of ephrin-A:EphA signaling on tumorigenesis, as well as tumor growth and spread has been extensively studied. This literature has been reviewed elsewhere [41,42] and will not be described in detail here. Briefly, aberrant expression of both EphAs and ephrin-As is found in a wide range of tumors. Ephrin-A signaling promotes or inhibits tumorigenicity depending on the cellular context and on whether the expression of both EphA and ephrin-A is elevated or reduced in the tumor. For instance, EphA2 signaling restrict the migration of glioma when activated by ephrin-A1, but in contrast, enhances cell motility when phosphorylated by the Akt kinase [43,44]. Another example lies in mutations in the EphA3 gene that have a high prevalence in lung adenocarcinoma [45].

## 3. Ephrin-A Forward Signaling Pathways

The widespread involvement of ephrin-A:EphA forward signaling in normal and pathological conditions led to the extensive investigation of intracellular events downstream EphA activation. Upon ephrin-A binding, EphAs undergo dimerization like most RTKs. This step promotes the trans-phosphorylation of each EphA intracellular domains, leading to EphA activation. This activation is driven mostly by a pair of anchored ephrin-As but can be achieved by its secreted monomeric counterpart. Cleaved by cell matrix metalloproteases, soluble monomeric ephrin-A1 is released from cancer cells and drives EphA2 dimerization and signaling [46,47,48]. The dimerization process relies on ligand binding as well as homotypic Eph-Eph contacts [49]. These contacts between neighboring Eph receptors involve several motifs located in the extracellular, intracellular, and transmembrane domains of EphAs [50,51,52]. The Ectodomains are highly conserved among Ephs. This homology is not restricted to EphAs but includes the related EphB family, allowing interclass EphA-EphB clusters [53]. Heterodimerization between EphAs and other RTKs members has also been observed (For review see [54]). Through intracellular domain interactions, FGF receptors can for instance dimerize with EphA4 and drive its activation [55]. Since some homotypic interactions are ligand-independent, EphA dimerization and activation can occur without ephrin-A binding. These ligand-free dimers have been observed for EphA2 and EphA3, respectively [52,56]. Even if ligand-free EphA3 dimers can have kinase activity, their capability to initiate EphA forward signaling is still debated. Already identified for other RTKs, dimerization without ligand binding might be used to prime ligand-dependent receptor activation [57]. Overall, EphA dimerization and activation can be accomplished through different modalities. Each dimerization modality might provide distinct signaling properties to EphA activity. This aspect might reflect their diversity of action [52].

EphAs can assemble in large clusters. Due to homotypic interaction, ligand-bound and ligand-free EphAs can be present. The precise composition and the size of EphA clusters are variable, adding another level of complexity in EphA signaling. The strength of the intracellular response depends on the size of these clusters.

Following EphA activation, the kinase activity of EphA and/or the recruitment of intracellular binding partners initiates downstream signaling pathways. Once EphA is activated, some trans-phosphorylated residues enable the docking of several interactors including Src family kinases (SFKs), members of the Ephexin family or Vav2/3. Ephexin-1, 3 and 5 are guanine nucleotide exchange factors (GEFs) that activates RhoA upon EphA binding to ephrin-As, leading to cytoskeletal reorganization [58,59,60,61,62]. Ephexin-4 is a RhoG GEF, that activates this RhoGTPase downstream of EphA2 in a ligand-independent way in breast cancer cells leading to enhanced cell migration [63]. Vav2 and 3 are GEFs that control Rac1, leading to filopodial formation and endocytosis of the receptors when EphAs are activated [24,64]. EphA10 diverges from the other member of the EphA family since it lacks kinase and docking properties. EphA10 is overexpressed in some cancer cells and its precise role in EphA signaling is still unclear [65,66].

Cyclic nucleotides (cAMP and cGMP) and calcium, are also key players downstream of ephrin-A:EphA forward signaling although their precise interactions with the ephrin-A:EphA pathway is still unclear. Cyclic adenosine 3’,5’-monophosphate (cAMP) and its downstream effector protein kinase A (PKA) modulate ephrin-A-induced morphological changes in developing and regenerating axons [67,68,69], in cholangiocytes [70] and in prostate cancer cells [71]. A reduction of cAMP concentration is induced by EphAs upon ephrin-A binding [72,73]. cAMP manipulation targeting either this second messenger directly [72], one of its synthesizing enzyme, adenylate cyclase 1 (AC1) [69,74,75] or its downstream effector PKA [69] are sufficient to impair the ephrin-A-induced retraction of retinal axons and to affect their connectivity. Cyclic guanosine 3’,5’-monophosphate (cGMP) is a signaling molecule closely related to cAMP. It is involved in ephrin-A:EphA forward signaling in hippocampal neurons through the activation of cGMP dependent kinase (PKG) [76], and is required for ephrin-A-induced pathfinding of retinal axons [77]. The ephrin-A:EphA pathway also involves calcium signaling for the downstream modulation of vascular cell adhesion molecule-1 (VCAM-1) in endothelial cells [78] and for morphological changes in axonal growth cones [68,79]. Preventing calcium influx into retinal ganglion cell axons, blocking the L-type calcium channels or impairing the calcium release from the intracellular stores alters ephrin-A-induced axon retraction, demonstrating that different sources of calcium interact with EphA downstream pathways [68]. 

The identification of the ligands (ephrin-As), receptors (EphAs) and downstream effectors (summarized in Figure 1) enabled the manipulation of the ephrin-A:EphA system at different levels by specifically targeting the different components of these signaling cascades. Available approaches range from pharmacological and peptide-based strategies to blocking antibody and genetically-encoded methods. The relevance of these approaches lies in usefulness to dissect the cellular functions and molecular pathways of ephrin-As in a healthy organism and their therapeutic potential to target EphA signaling in pathological conditions.

## 4. Targeting EphA Receptors

Within the toolset available to interfere with ephrin-A:EphA forward signaling, many strategies target EphA receptors. Antibodies, interfering RNAs (iRNAs), peptides, soluble fragments of Eph/ephrin, and small molecules—including protein-protein interaction inhibitor (PPIs) and kinase inhibitors—have been used. Since reviews summarizing therapeutic-oriented strategies are available in the literature [6,80,81,82], we will here focus on recently developed tools targeting EphA receptors.

### 4.1. Antibodies

Developing monoclonal antibodies (mAbs) is a common approach to design cancer therapies [83]. mAb targeting oncogenes are used to modulate the immune system or to interfere with signaling pathways. They exhibit antagonist or agonist effects on their target. In competition with the native ligand, antagonist mAbs block the signaling pathway by fixing with high affinity the ligand binding domain (LBD) of the receptor. Since EphA expression is altered in many tumors, mAbs targeting these receptors have been developed. For instance, DS-8895a, an anti-EphA2 mAb, inhibits the phosphorylation of EphA2 in cancer cell lines exposed to ephrin-A1 [84]. Most of anti-EphA mAbs available exhibit an agonist effect. Agonist mAbs activate the signaling pathway either by mimicking the native ligand activity or by stabilizing or driving receptor oligomerization. SHM16, an anti-EphA2 mAb, mimics the anti-oncogenic effect of ephrin-A1 on melanoma cell line [85]. Another anti-EphA2 mAb, 1C1 (also called MEDI547 when conjugated to cytotoxic agent), triggers the tyrosine phosphorylation of EphA2, leading to its internalization and degradation [86]. IIIA4, an anti-EphA3 mAb (also called KB004 or Ifabotuzamab) recognizes a specific conformation of EphA3 and facilitates the assembly of ephrin-A5:EphA3 clusters [87]. The mechanisms underlying the activation of the targeted receptor by agonist mAbs are complex and depend on its biophysical features, including binding epitope, affinity, or receptor occupancy (for review see [88]). 

Another approach to interfere with EphA signaling is to block the ephrin-A:EphA interaction by targeting the ligand itself. This strategy has been sparsely used. Nevertheless, PF-06647263 an anti-ephrin-A4 conjugated with a cytotoxic agent has been developed and exhibits a tumor suppressor activity [89].

Besides mAbs, other strategies like nanobodies and single-chain antibodies (scFvs) have been developed and are likely to offer new ways to target EphAs. Nanobodies are an antigen-binding fragment derived from variable regions of the heavy-chain antibody of camelids, a group of species whose antibodies lack light chains [90]. scFvs consist of the variable domain of light and heavy chain of an antibody fused together with a flexible peptide linker [91]. Both are more stable and easier to generate than mAbs. Like mAbs, nanobodies and scFvs exhibit either agonist or antagonist effects. For instance, two anti-EphA4 nanobodies (Nb39 and Nb53) antagonize the downstream signaling of this receptor by blocking its phosphorylation [92]. Due to their small size (15 and 30 kDa respectively), they enable improved tissue penetration. A nanobody against GFAP, an intermediate filament protein, has been found to cross the blood-brain barrier in vivo and to bind its intracellular target [93]. This might provide a new way to modulate EphA signaling, by targeting its intracellular domains, including the kinase motif, although to date, this approach has not been implemented for EphA receptors. Finally, nanobodies and scFv can be fused together, creating multimeric nanobodies or scFvs with the ability to recognize multiple epitopes from the same target or from multiple targets for combinatorial effects [94,95]. A bispecific anti-EphA2/EphA3 antibody triggers internalization of EphA2 and EphA3 leading to reduced tumorigenicity of glioblastoma [96]. Since Eph:Eph interactions are crucial for the formation of receptor clusters and the activation of downstream signaling [49,97], those tools might provide promising strategies for acute modulation of EphA signaling. 

Overall, the use of antibody-based strategies enables the manipulation of EphA signaling with some specificity for each member of this receptor tyrosine kinase family.

### 4.2. Interfering RNAs

MicroRNAs (miRNAs) are endogenous non-coding RNAs inhibiting the expression of specific genes by binding partially to the corresponding messenger RNAs (mRNAs). Based on this mechanism, two mains approaches have been designed to modulate gene expression: blocking specific miRNA or mimicking their effects with non-endogenous interfering RNAs (iRNAs). miRNA inhibitors (also called antagomiR) are antisense oligonucleotides that through their inhibiting process promote overexpression of genes regulated by the targeted endogenous miRNA. For instance, miR-145 inhibitor leads to the overexpression of EphA4 in human cortical neuron [98] and miR-26a inhibitor enhances EphA2 expression in endothelial cells [99]. Since miRNAs control the expression of multiple genes, the impact of miRNA inhibitors is not restricted to EphA genes. Besides EphA4 and EphA2, miR-145 and miR-26a modulate the expression of Twist1 in human colorectal cancer cells [100] and the expression of TRPC3 in human aortic endothelial cells [101] respectively.

Non-endogenous iRNAs are divided into 3 main subtypes: miRNA mimics, small interfering RNAs (siRNAs), and short-hairpin RNAs (shRNAs). miRNA mimics have the same sequence as endogenous miRNAs. Therefore, they exhibit the same lack of specificity for the targeted mRNA. For instance, a miR-124 mimic was used to demonstrate the involvement of EphA2 in chemotherapeutic drug resistance in pancreatic cancer cells [102] but also modulates the expression of BACE1, a protease involved in amyloid precursor protein cleavage, a process that underlies the pathogenesis of Alzheimer’s disease [103]. Unlike miRNA mimics, siRNAs and shRNAs exhibit higher specificity by binding specifically the targeted mRNA. siRNAs against EphA4 have been used to highlight the negative role of EphA4 in recovery after brain hemorrhages [104] while an shRNA against EphA7 was employed to demonstrate its implication in the ovulation process [105]. Despite their common higher specificity, siRNAs and shRNAs present several differences (for review see [106]). Even if siRNAs are oligonucleotides carefully designed to be fully complementary to a specific mRNA, they share the same physicochemical properties with miRNA mimics. Therefore, they have a short half-life and do not penetrate the plasma membrane easily *in vivo*. In contrast, shRNAs are delivered into the cell in a DNA plasmid form. This approach makes them more stable than other iRNAs and provides a longer gene silencing effect. However, the control of their expression level is challenging, their design is not straight forward, and they induce a higher cellular toxicity. 

Lots of new carrying methods for iRNAs delivery have emerged in the past decade in order to enhance their efficiency *in vivo*. Nanocarriers can be distinguished into viral and non-viral vectors. As mentioned above, viral vectors are used for shRNAs but also for miRNA mimics/inhibitors against EphAs [107,108]. Even though the use of viral vectors is an effective way to deliver iRNAs *in vivo*, their high production cost, together with their immunogenicity and the risk of insertional mutagenesis make them less attractive for therapeutic strategies. Consequently, a broad diversity of non-viral nanocarriers has been generated. They can be made of lipids, polymers, metal, mesoporous silica or porous silicon (for review see [109]). Those carriers are mostly used for siRNAs. For instance, a cholesterol-modified siRNA against EphA4 was used to enhance axon regeneration in rats [110]. siRNAs embedded in silicon porous particles, called SIMPs, have been used to provide EphA2 silencing over 3 weeks in a mouse model of ovarian cancer [111]. Additionally, they can be addressed in vivo to a specific cell population by expressing on their surface antibodies or peptides [109].

Supported by their new non-viral nanocarrier delivery strategies, their high specificity and low production cost, siRNAs appear as a method of choice to interfere with EphA expression and provide an interesting approach to interfere with EphA signaling when other strategies are missing. However, iRNAs are more indicated for sustained EphA modulations rather than acute alterations of their downstream pathways.

### 4.3. Peptides and Soluble EphA/Ephrin-A

Peptides designed to interfere with EphAs share common features with antibody-based approaches including a high affinity and high selectivity. Their small size enables enhanced tissue penetration in vivo and their production is more cost-effective than antibodies. However, their stability is reduced, and their pharmacokinetics and bioavailability in vivo are low. Agonist and antagonist peptides are available. Both classes often interact with the extracellular LBD domain of EphAs. For instance, two peptides called YSA and SWL trigger EphA2 tyrosine phosphorylation and its downstream pathways [112]. In contrast, KYL, VTM, and APY peptides compete with the native ligand and prevent EphA4 activation [113]. An extensive description of the available EphA-targeting peptides is available elsewhere [114].

An advantage of peptides is their easy chemical modification that leads to optimized affinity or enhanced agonist potency. For instance, βA-WLA-YRPK-bio, a YSA/SWL modified peptide that is biotinylated at its C-terminal end and contains an additional β-alanine in N-terminus, exhibits both a higher affinity and a higher ability to induce EphA2 signaling due to its strengthened EphA clustering effect [115]. Since the size of EphA clusters controls the strength of the downstream pathway, DNA trimeric nanostructures have been used to couple three non-modified SWL peptides [116]. Those DNA-based structures serve as a scaffold to spatially gather conjugated ligands and facilitate receptor oligomerization [117]. Another strategy with a similar rational has been developed using homodimeric modified peptides enabling covalent binding of two peptidic units. YSA-derived 123B9 and 135H11 peptides used in a dimeric form induce EphA2 activation when used at a lower concentration than their monomeric counterpart [118,119]. Finally, chemical peptide modifications enable the development of antagonist peptidic chains from agonist ones by changing their local charges. Removing the C-terminal negative charge of the YSA agonist derivative βA-WLA-YRPK enhances its affinity and suppress its ability to activate EphA2 signaling, turning it into a competitive antagonist [115].

In addition to peptides interacting with EphA LBD, novel strategies targeting other domains of EphAs have emerged. A remarkable agonist peptide named TYPE7 can for example interact with the transmembrane domain of EphA2. TYPE7 is a pH-sensitive peptide derived from the transmembrane domain of its target. Inducing a pH drop leads to the translocation of the peptide to the plasma membrane where it binds EphA2, promotes its oligomerization, and activates its downstream signaling [120]. A subset of peptides also targets EphA intracellular domains including the Sterile Alpha Motif (SAM) of EphA2. EphA2 SAM domain is able to bind similar motifs from other proteins, including the lipid phosphatase Ship2 [121]. This heterotypic SAM:SAM interaction negatively modulates EphA2 endocytosis [122]. A peptide termed ShipH1 was designed to antagonize EphA2:Ship2 interaction [123]. However, even though the interaction between ShipH1 and EphA2-SAM has been validated *in vitro*, the specificity of this peptide, together with its ability to cross the cell membrane and interfere with EphA2 signaling need to be further characterized. Targeting an intracellular domain of EphA2 using a peptidomimetic molecule also enables to inhibit EphA2 phosphorylation in vitro [124]. Peptidomimetics are biomimetic oligomers able to mirror protein and peptide structures. Their unnatural backbones make them more stable than peptides (for review see [125]). Cyclic peptidomimetics like the one targeting EphA2 exhibit an improved membrane permeability compared to amino acid chains (for review see [126]), opening new ways to modulate EphA signaling through intracellular manipulation. 

Soluble fragments of EphAs and ephrin-As remain a common method to modulate EphA forward signaling. EphA soluble fragments are chimeras of the extracellular domain of EphAs and the Fc domain of IgGs to enhance their stability. They antagonize EphA forward signaling by competing with endogenous EphAs for ligand binding. Blocking EphA6 signaling with EphA6-Fc limits cell death after brain injuries [127]. However, since they bind ephrin-As, they can also stimulate the reverse signaling pathway [128]. This bimodal action of EphA-Fc complicates the conclusions drawn when using this approach. The same chimeric approach for ephrin-As is available to promote EphA activation [127,129]. However, this method also prevents reverse signaling by competing with endogenous ephrin-As. Both EphA-Fc and ephrin-A-Fc are often used in a multimeric form to improve their antagonist/agonist effect. Clustered EphA-Fc or clustered ephrin-A-Fc are usually created by using anti-IgG (Fc) antibody [130], but the DNA nanostructures strategy described above for peptides was initially developed to cluster recombinant ephrin-A5 [117]. 

Peptides are an attractive approach to control EphA forward signaling considering their high selectivity and potential of chemical modifications, whereas soluble fragments of EphA or ephrin-A mimic closely endogenous signaling. Taking together, both peptides and soluble fragments represent an abundant toolset to control ephrin-A:EphA pathways. 

### 4.4. Small Molecules

Small molecules targeting EphAs are classified into two classes that target EphAs through distinct mechanisms: protein-protein interaction (PPI) inhibitors that prevent ephrin-A:EphA binding and molecules that control the kinase activity of EphAs. 

PPI inhibitors target the LBD motif with high affinity, although lower than peptides and antibodies. They are more cost-efficient than their peptidic and antibody counterparts but are also less effective. However, the extensive library of available chemical compounds led to the identification of a large set of PPI inhibitors as potential EphA signaling modulators. Like peptides, recent efforts have been focused on optimizing existing PPI inhibitors since most of them exhibit lack of specificity and stability, as well as weak bioavailability [131]. PPI inhibitors targeting EphAs have been reviewed recently [82]. Recent improvements are summarized here. UniPR1331 and UniPR500 are two derivatives of Lithocholic acid (LCA), an EphA2 antagonist [132]. Both of them exhibit an enhanced bioavailability and inhibit EphA2 and EphA5 signaling respectively [133,134,135]. Like LCA, UniPR1331 inhibition occurs in a dose-dependent manner and UniPR500 blocks both forward and reverse Eph signaling. However, those derivatives lack specificity and bind all Eph receptors, suggesting that they might interact with a cavity within the LBD domain among all Ephs. In addition to PPI inhibitors, a single small molecule derived from Doxazosin was reported to elicit agonist EphA activity. Doxazosin is an α1-adrenoreceptor antagonist that also activates EphA2/EphA4 signaling by docking their LBD [136]. A modified dimeric form of Doxazosin provides a stronger activation of EphA2 signaling [137]. Despite a residual affinity for α1-adrenoreceptor, this compound presents a higher specificity and affinity for EphA2 than its parent inhibitor and is relatively stable *in vivo*, encouraging future investigations for therapeutic prospect [138].

According to their precise docking site and inhibition modalities, kinase inhibitors (KIs) are divided into five classes. All KI families but type III interact with the ATP-binding site of the kinase (for review see [139]). Designing KIs for EphAs has been challenging since the ATP-binding domain is highly conserved among RTKs. Thus, several non-specific KIs including INNO-406 or Dasatinib have already been used to prevent EphAs signaling [140,141]. Nevertheless, several attempts aimed at improving their specificity. Chemical modifications of Dasatinib have reduced the number of its known targets from 44 to 31 while keeping its high affinity for EphA2 [142]. Similarly, a few optimized KIs with enhanced specificity for EphA3 have been developed, however still with cross-reactivity with many other kinases [143]. 

Despite the extensive work done for the last decades, small molecules still appear as a strategy lacking specificity compared to other available approaches. This limitation is more marked for KIs that hit other kinases compared to PPI inhibitors that exhibit specificity for Eph kinases without the ability to discriminate between the members of this family.

Overall, the diversity of usable tools (summarized in Table 1) together with the easy accessibility of the extracellular domain of EphAs point towards such approaches to manipulate EphA forward signaling. 

## 5. Targeting RhoGEFs

### 5.1. Ephexins

Guanine nucleotide exchange factors (GEFs) are key effectors of the cellular cascade induced by EphA forward signaling. They activate Rho GTPases downstream of EphAs by enhancing the exchange from GDP- to GTP-bound, thus switching Rho GTPases from an inactive to an active state [144]. Ephexin-1, a member of the GEF family is an interesting target to manipulate ephrin-A forward signaling since it is a direct interactor of EphA cytoplasmic domain [58] and its expression pattern is restricted to the central nervous system where EphAs have a variety of roles during development. Thus, manipulating Ephexin-1 enables to specifically target a subset of cells expressing EphAs. Ephexin-1 plays an important role in opposite cellular processes modulated by EphA activation. In absence of ephrin-A, Ephexin-1 activates RhoA, Rac1, and Cdc42 and promotes axon outgrowth. Ephrin-A binding to EphAs induces Src-dependent phosphorylation of Ephexin-1 and enhances RhoA activation without affecting the activity of Rac1 and Cdc42, changing the balance between the downstream pathways of these Rho GTPases. It in turn promotes ephrin-A-induced actin depolymerization and axonal growth cone repulsion [59]. Among the different strategies to target this protein, to the use of interfering RNAs enables to target directly Ephexin-1. This technique was successfully used in chicken, by using shRNA against c-ephexin, the endogenous chicken ortholog of Ephexin-1. Expressing Ephexin-1-targeting shRNAs in vitro in HEK293T cells and in vivo in hindlimb motor neurons in the spinal cord reduces c-ephexin expression and leads to the premature entry of motor axons into the hindlimb of chick embryos [59]. Like the strategies targeting Ephexin-1, approaches to manipulate Ephexin-3, 4 and 5 are limited to shRNA- or siRNA-based methods. siRNAs targeting Ephexin-3 have been used to highlight its role in the Src-induced podosome formation [145], whereas an shRNA-induced knock-down of Ephexin-3 led to the identification of its role in the migration of dendritic cells [146]. An Ephexin-4 shRNA enabled the identification of a RhoG-induced pathway downstream of the EphA2:Ephexin-4 interaction [63]. Knocking-down Ephexin-5 using an shRNA protects a mouse model of Alzheimer’s disease from developing cognitive impairment [147]. However Ephexin-5 manipulations are not specific to EphA signaling since EphBs are also placed upstream of this GEF [148].

### 5.2. Vav2 and 3

Among the GEF family, Vav2 and Vav3 are critical actors for ephrin-A-induced axon repulsion. Like Ephexin-1, Vav2 and Vav3 are recruited to the cytoplasmic domain of EphAs upon ephrin-A:EphA binding [24]. Once bound to EphAs, they are activated and in turn enhance Rac1 activity, leading to the Rac-dependent endocytosis of EphA:ephrin-A complexes. Endocytosis of these EphA assemblies is necessary to switch from the initial cell-cell adhesion required for the binding of ephrin-As to EphAs, to repulsion [24,64]. In contrast to Ephexin-1, -3 and -4 that binds specifically to EphAs, Vav proteins also transduce signaling from other receptors including EphBs [64], making them targets of lower specificity to manipulate the ephrin-A:EphA forward signaling.

A direct way to inhibit Vav2/3 is to reduce their expression using interfering RNAs. This technique enables to deplete Vav2/3 expression in a variety of cell types and to inhibit its modulation of a range of physiological processes. SiRNA against Vav2 and Vav3 in PC12 cells is sufficient to inhibit the activation of Rac1 and Cdc42 [149]. In Schwann cells, the expression of a Vav2 siRNA prevents the ephrin-induced reduction of cellular migration [150]. siRNA-induced knockdown of Vav2 in INS-1 832/13 beta cells inhibits Rac1 activation and glucose-stimulated insulin secretion [151].

Altering ephrin-A forward signaling through Vav2 or 3 manipulation is also possible by inhibiting the formation of the Vav2/Rac1 complex. This approach can be implemented by using the GTPase inhibitor EHop-016 [151,152,153]. However, this compound targets the surface of Rac1 to inhibit Vav2 binding and is thus unlikely to be Vav2-specific. EHop-016 might also block the binding of other Rac1-targeting GEFs including Tiam1 [152].

Vav proteins are composed of different structural motifs: a Dbl homology (DH) domain, a calponin homology (CH) domain, a C-terminal pleckstrin homology (PH) domain, and a C1 domain. Recently, the C1 domain of Vav proteins and of Vav3 in particular, has been identified as a potential therapeutic target to inhibit the members of this family [154,155]. This Vav fragment is an atypical C1 domain with close homology to the one of protein kinase C. But in contrast to the latter, the C1 motif of Vavs is not able to bind phorbol esters [154]. This specificity has been used to disrupt the signaling cascade downstream of Vav3 by expressing a modified Vav3 with a C1 domain able to bind phorbol ester that in turn acts as a dominant-negative form. Expression of a variant of Vav3 with this modified C1 domain leads to a change of the Vav3 localization, now found at the plasma membrane instead of the cytoplasm, leading to a disruption of the interactions with its partners [155].

Vav functions rely on different allosteric changes. The binding to GTPases induces a conformational change in the GTPase-binding DH domain of Vavs, altering their activity. Vav activity is also regulated by an intramolecular interaction with its autoinhibitory domain (AID). A toolset enabling the controlling of these allosteric switches with light or ligands is available to manipulate Vav2 activity [156,157]. Allosteric control of Vav2 can be triggered by insertion of domains like uniRapR (rapamycin-controlled) and LOV2 (light-activatable) in appropriate allosteric sites. Successful activation of Vav2 can be triggered by the insertion of LOV2 in a loop of the AID domain, allowing the photo-activation of the chimeric protein. The insertion of the uniRapR domain inside the DH motif enables the activation of the synthetic protein when rapamycin is present. The opposite manipulation (Vav2 inhibition) has been achieved by the insertion of a LOV2 motif in the DH domain, enabling photo-inhibition of Vav2 [156].

In conclusion, a variety of approaches (summarized in Table 1) are available to target GEFs downstream of the ephrin-A:EphA forward pathways. They enable the manipulation of only a subset of the EphA downstream pathways, targeting either the Ephexin- or Vav-dependent signaling. However, all strategies modulating Vav2/3 and Ephexin suffer from a lack of specificity since these EphA effectors themselves are involved in other pathways including EphBs for Vav2/3 and Ephexin-5.

## 6. Targeting cAMP, cGMP, and Calcium

The modulation of second messengers, including cyclic nucleotides (cAMP and cGMP) and calcium, is required for ephrin-A:EphA forward signaling. They are also critical for a large diversity of cellular mechanisms, leading general approaches targeting second messengers to lack specificity for the EphA pathway. However, endogenous second messenger signals achieve specificity for each of their downstream signaling cascades. The precise subcellular localization of their signals is a crucial component of this specificity. The cellular compartments of cAMP signals involved in transducing EphA downstream cascades is being elucidated and highlights lipid rafts, a set of discrete and highly dynamic subdomains of the plasma membrane enriched in sphingolipids and cholesterol [72,77,79]. The cAMP, cGMP, and calcium signals required for ephrin-A-induced axon retraction are restricted to this plasma membrane compartment [72,77,79]. Controlling second messenger concentration in this cellular domain provides approaches to manipulate the ephrin-A:EphA pathway with enhanced specificity compared to altering cAMP, cGMP, and calcium in the entire cell.

### 6.1. cAMP

Ephrin-A signaling is one among many pathways controlled by cAMP leading to the lack of specificity for most of tools enabling the manipulation of this second messenger. This applies to the pharmacological strategies controlling the activity of adenylyl cyclases (the cAMP synthesizing enzymes) including Forskolin [158] and SQ22536 [159] that stimulate and inhibit transmembrane adenylyl cyclases respectively. The same limitation stands for general inhibitors of phosphodiesterases, the cyclic nucleotide hydrolyzing enzymes including IBMX. These pharmacological agents have been used to demonstrate the involvement of cAMP in the EphA downstream pathway [68,69,71] but lack specificity for this signaling cascade.

Recently developed approaches enable subcellular specific manipulations of cAMP restricted to lipid rafts where the modulations of this second messenger transduce EphA downstream signaling [72,73]. Such manipulations are based on genetically-encoded tools, including a synthetic cAMP scavenger and a light-driven adenylyl cyclase. cAMP Sponge is a dominant-negative form of the cAMP downstream effector PKA [160]. It binds cAMP with a high-affinity and act as a scavenger of this second messenger, thus providing inhibition of its downstream signaling. Since cAMP Sponge is genetically encoded, it has been fused to a short targeting sequence derived from Lyn kinase, restricting its expression to lipid raft to manipulate ephrin-A forward signaling and limit the impact of this manipulation on other cAMP-dependent signaling pathways [72]. A similar approach has been developed using the light-sensitive adenylyl cyclase bPAC [161]. Using the same Lyn kinase-derived targeting sequencing, restricting bPAC to lipid rafts enables to manipulate cAMP concentration exclusively in this cellular compartment. Lipid raft targeted bPAC has been used to mimic the ephrin-A-induced retraction of developing axons [72].

### 6.2. cGMP

Like for cAMP, approaches enabling global cGMP manipulation face specificity issues for the majority of tools enabling to manipulate cGMP. This limitation stands for the pharmacological strategies changing the activity of guanylyl cyclases, the cGMP synthetizing enzymes, or manipulating the degradation of cGMP with phosphodiesterase inhibitors.

To circumvolve the limitations of manipulation of cGMP in the entire cell and increase the specificity for the ephrin-A:EphA pathway, approaches aiming at controlling this second messenger in subcellular domains have been developed. This includes a genetically-encoded cGMP scavenger termed SponGee and based on the cGMP binding site of a high affinity variant of the cGMP effector PKG [77]. This design enables to limit its expression and thus the manipulation of cGMP signaling to a biochemically-defined subcellular compartment using targeting sequences. Targeting SponGee to lipid rafts with the short targeting sequence deriving from the Lyn-kinase allows to manipulate ephrin-A forward signaling and to limit the effect of the manipulation on other cGMP-dependent pathways. Blocking cGMP inside lipid rafts enables to prevent ephrin-A5-induced retinal growth cone collapse whereas excluding it from lipid rafts does not [77].

### 6.3. Calcium

Like strategies targeting cAMP and cGMP, the general approaches developed to manipulate calcium lack specificity against the ephrin-A pathway and is a serious limitation of the use of most calcium manipulation strategies. It applies to the synthetic calcium buffers (e.g., EGTA [162]) or the large variety of calcium channels inhibitors [163]. Both manipulations prevent the ephrin-A-induced retraction of axons [68], like the depletion of intracellular calcium stores using thapsigargin [68].

As described previously, calcium achieves specificity for its different signaling pathways by strategies including spatial restriction of its signals in cellular microdomains. Calcium manipulation with subcellular resolution is available using SpiCee, a genetically encoded calcium buffer derived from calmodulin and parvalbumin. SpiCee enables lipid raft-restricted manipulation of calcium, a set of the calcium signals involved in ephrin-A signaling in developing axons [79]. This strategy enables to enhance calcium manipulation specificity for ephrinA downstream pathways.

In conclusion, there is a wide range of approaches available to manipulate second messengers (summarized in Table 1). However, these techniques often lack specificity for the ephrin-A pathway. The recent development of subcellular-specific variants of genetically-encoded tools enables to restrict the manipulation of second messengers in cellular microdomains associated with ephrin-A:EphA forward signaling, thus increasing their selectivity. This observation stands for three ubiquitous second messenger involved in the EphA transduction cascade: cAMP, cGMP, and calcium.

## 7. Conclusions

Manipulating ephrin-A:EphA signaling can be achieved by a wide range of approaches and tools summarized in Figure 2 and in further details in Table 1. Whereas antibody- and peptide-based strategies targeting EphA-receptors are likely the most specific approaches available, they do not enable to differentiate between the distinct downstream pathways of these receptors. The specific control of direct EphA effector activity s (Ephexins and Vavs) offers this possibility at the expense of a decreased specificity for ephrin-A forward signaling. Modulating second messenger signaling involved in the EphA pathway using recently developed tools enabling subcellular-specific control of these signaling molecules offers alternative strategies with an enhanced specificity for ephrin-A signaling than the alteration of the cAMP, cGMP, or calcium signaling spread over the whole cell. Further understanding of the second messenger signals controlling EphA signaling might pave the way to further specificity for cyclic nucleotide- and calcium-approaches.

## Figures and Tables

**Figure 1 pharmaceuticals-13-00140-f001:**
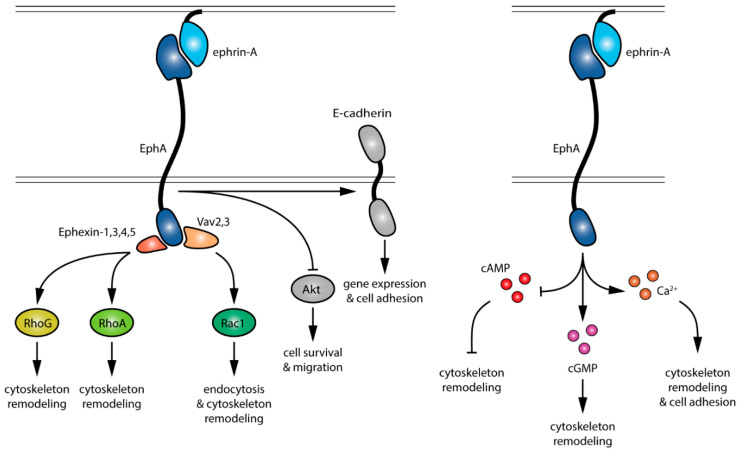
Signaling events downstream of EphA activation by ephrin-As. Pathways in gray are not detailed in this review.

**Figure 2 pharmaceuticals-13-00140-f002:**
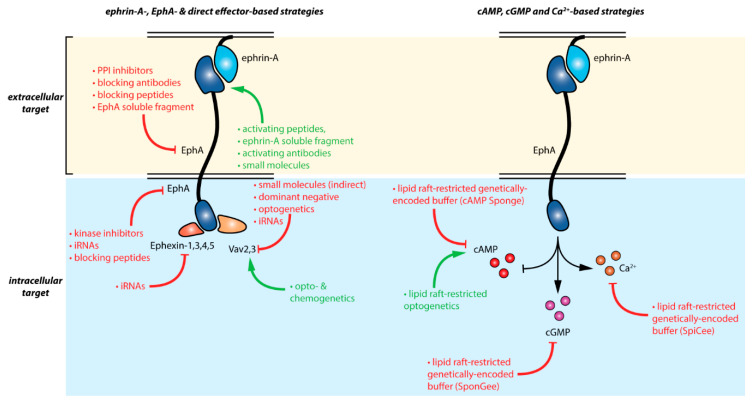
Summary of the available strategies enabling the manipulation of ephrin-A:EphA signaling. Red and green arrows indicate inhibitor and activator effects respectively. Details are provided in Table 1.

**Table 1 pharmaceuticals-13-00140-t001:** Summary of available approaches to manipulate ephrin-A:EphA forward signaling. The mechanism of activating and inhibiting approaches is denoted in green and red respectively.

ephrin-A:EphA Effectors	Technique	Tool	Targets	Mechanisms	Advantages	Limits	References
EphA Targeting							
**EphA/ephrin-A**	Antibodies	mAbs	LBD extracellular domain in a given conformation endogenous ligand	Mimic endogenous ligands Drive EphA oligomerization Compete with endogenous ligands	High affinity, stability & selectivity Multiple epitope targeting (nanobodies and scFvs) Efficient tissue penetration in vivo (nanobodies and scFvs)	Lack access to intracellular signaling Costly	[84,86,87,89,92,96]
		Nanobodies					
		scFvs					
	iRNAs	miRNA inhibitor	miRNA controlling EphAs expression	Inhibition of miRNA expression (leading to overexpression of the targeted EphA)	Straightforward implementation Cost-effective	Low specificity Requires intracellular delivery	[98,99,100,101]
		miRNA mimics	EphA mRNA	Inhibition of EphA expression	Straightforward implementation• Cost-effective Cost-effective In vivo delivery possible if using nanocarriers High specificity (siRNA) High stability (shRNA)	Low specificity (miRNA mimics) Limited to chronic manipulations Requires intracellular DNA or RNA delivery Risf of off-target effects (ShRNA)	[102,103,104,105,108,111]
		siRNA					
		shRNA					
	Peptides and soluble fragments of ephrin-As or EphAs	Peptides	LBD Transmembrane domain Intracellular domains	Mimic endogenous ligand activity Drive receptor oligomerization Compete with endogenous ligands	Cost-effective High selectivity and good affinity Diversity of mechanism Efficient tissue penetration Enhanced membrane permeability (modified peptide)	Low stability Low pharmacokinetics and bioavailability *in vivo*	[112,113,114,115,116,118,119,120,123,124]
		EphA-Fc	endogenous ephrin-As	Compete with endogenous EphAs for ligand fixation	High affinity/stability/selectivity Can be fused to target multiple epitopes	Costly Can interfere with reverse signaling	[117,127,129,130]
		Ephrin-A-Fc	endogenous EphAs	Mimic endogenous ligands			
	Small molecules	PPI inhibitors	LBD	Compete with endogenous ligands	Cost-effective Specific for EphAs	Low specificity within the EphA family Can interfere with reverse signaling	[132,133,134,135,136,137,138]
		Agonists		Mimic endogenous ligands			
		Kinase inhibitors	Intracellular ATP-binding site	Inhibition of EphA catalytic activity	Can interfere with catalytic activity of EphAs	Low specificity for EphAs	[140,141,142,143]
**RhoGEF Targeting**							
**Ephexins**	iRNAs	shRNA, siRNA	Ephexin mRNA	Inhibition of Ephexin expression	Sustained inhibition	Risk of off-target effects Limited to chronic manipulations Requires intracellular DNA or RNA delivery	[59,63,145,146,147]
**Vav2/3**	iRNAs	siRNA	Vav2 or Vav3 mRNA	Inhibition of Vav2 or Vav3 expression	Direct Vav2 or 3 targeting	Limited to chronic manipulations Requires intracellular RNA delivery	[149,150,151]
	Small molecules	Ehop-016	Rac1	Inhibition of Rac1:Vav2 interaction	Straightforward cell delivery	Low specificity (might also block Rac1 binding to other GEFs)	[151,152,153]
	Protein variants	Vav3 with modified C1 domain	Vav3	Alter Vav3 subcellular localization	Direct Vav3 targeting	Requires intracellular DNA delivery	[154,155]
	Control of allosteric switches	LOV2-based strategies	AID domain/DH domain	Activation of Vav2 activity Inhibition of Vav2 activity	Direct control of Vav activity High temporal resolution	Requires intracellular DNA delivery Does not control the activity of endogenous Vavs Restricted to acute manipulations Limited use in vivo (requires optical access)	[156,157]
		uniRapR	DH domain	Activation of Vav2 activity	Direct control of Vav activity Temporal control of Vav activity Applicable *in vivo*	Requires intracellular DNA delivery Does not control the activity of endogenous Vavs	[156,157]
**cAMP, cGMP, and calcium Targeting**							
**cAMP**	Genetically-encoded tools	Light-activated adenylyl cyclases	cAMP	Synthesize cAMP	Temporal control Enhanced specificity for the ephrin-A pathway when targeted to lipid rafts compared to global cAMP manipulation	Requires intracellular DNA delivery Limited use in vivo or in thick tissue (requires optical access) Low specificity for EphA signaling	[72,161]
		cAMP sponge	cAMP	Buffer cAMP, inhibit downstream signaling	Enhanced specificity for the ephrin-A pathway when targeted to lipid rafts compared to global cAMP manipulation Enables long term manipulations	Requires intracellular DNA delivery Limited to chronic manipulations Low specificity for EphA signaling	[72,160]
**cGMP**	Genetically-encoded tools	SponGee	cGMP	Buffer cGMP, inhibit downstream signaling	Enhanced specificity for the ephrin-A pathway when targeted to lipid rafts compared to global cGMP manipulation Enables long term manipulations	Requires intracellular DNA delivery Limited to chronic manipulations Low specificity for EphA signaling	[77]
**Calcium**	Genetically-encoded tools	SpiCee	calcium	Buffer calcium, inhibits downstream signaling	Enhanced specificity for the ephrin-A pathway when targeted to lipid rafts compared to global calcium manipulation Enables long term manipulations	Requires intracellular DNA delivery Limited to chronic manipulations Low specificity for EphA signaling	[79]

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
