# Peer review of "Approaches to Manipulate Ephrin-A:EphA Forward Signaling Pathway"

_pharmaceuticals, 2020, doi:10.3390/ph13070140_

Round 1

Reviewer 1 Report

The review deals with the manipulation of EphA forward signaling through the use of multiple approaches.

Overall, the paper is unbalanced through aspecific and questionable targeting approaches and the chapter dealing with the signaling pathways is uncomplete and misleading .In fact minor pathways of Eph/ephrin signaling are represented while major pathways are missing. This representation leads the reader to think that you can manipulate calcium, cAMP and cGMP levels to finally modulate EphA signaling. Whilst, obviously, calcium , cAMP and cGMP levels are regulated by many physiological systems and are downstream to many effectors mediating a large number of actions. Ephrins play a minor role only in this story.

Accordingly, targeting approaches on cAMP, cGMP and calcium, which are highly aspecific, not selective and promiscuous must be removed.

IN DETAIL:

The title is somehow misleading because forward signaling is triggered by Eph kinase activation whereas reverse signaling is due to ephrins.

The second paragraph “ EphrinA- forward signalling” is incomplete and does not describe the following:

  • The activation of the receptor through ephrin soluble ligands,
  • The activation via “cis” interaction
  • The activation through heterodimerization with other RTKs (i.e. EGFR or ErbB2)
  • The Kinase dead receptor EphA10

Moreover, even if the Authors support the idea of cAMP,calcium or ephexin targeting, they should describe the more prominent second messengers involved in EPh forward signaling in this section. Figure 1 should be revised accordingly.

PAg 4 lines 4-6 The following sentence “Within the toolset available to interfere with ephrin-A:EphA forward signaling, many strategies target EphA receptors. This abundance might reflect the higher accessibility of EphAs to drugs compared to their intracellular downstream effectors.” is unclear, it must be revised and better explained. It looks like the Authors would suggest a connection between the accessibility of the binding site and the downstream effectors. However, I am not aware of such a kind of correlation.

Moreover, the presence of kinase inhibitors is not surprising since the catalytic domain of kinases is highly conserved and kinase inhibitors are highly promiscuous blocking kinase activity of a large number of kinases. The presence of biological (siRNA, antibodies) is not even surprising since you can project a siRNA for any protein in our body.

The real surprise is the possibility of inhibiting Eph/ephrin interaction with small molecules. In fact protein/protein interaction is usually mediated through the interaction of large surfaces of the receptor and the ligand making the small molecules useless. On the contrary Eph/ephrin interaction needs the insertion of ephrin G-H loop in the binding site of the Eph receptor. 

ANTIBODIES:

PF-06647263 and MEDI547 should be included

iRNAs

This long paragraph is mainly a general overview on iRNAs and I am not sure it is useful to include it in the paper. Probably, I would summarize it by directly explaining their in vivo delivery drawbacks

Small molecules

Pag 8 line 31: The sentence:” Despite the extensive work done for the last decades, small molecules still appear as a strategy lacking specificity” is correct when talking about kinase inhbitors but it must be corrected when talking of PPI-inhibitors. Even if small molecules PPI-i are unable to discriminate between Eph kinases classes the do not interact with other kinases or target. I suggest to change with ” Despite the extensive work done for the last decades, kinase inhibitors still appear as a strategy lacking specificity

Ephexin-1

The regulation of ephexins could be an interesting approach in Eph targeting. However, it is unclear why EPhexin-1 only was included in the paper when Ephexin3 has been shown to interact with EphA4, ephexin4 with EphA2 and ephexin5 with EphA4. I suggest to include them in the paper.

Pag 15 line 20: targeting src is far to be a specific approach since src regulates many other proteins. Consequently, the use of the suggested src inhibitors will result in the modulation of many other target other the EPh/ephrin

Pag 15 line 24: PP2 is not a specific src inhibitor since it blocks the activity of many kinase at similar concentrations (PMID: 22594480)

Pag 15 line 27 Since Authors state that “…these approaches are neither specific for Ephexin-1 nor for Src among the SFK family” I suggest to widely revise lines 21-26 in a more consistent way

Vav2 and vav3

They are downstream of many kinases since the can not be regarded as an approach to modulate ephrin-A forward signaling. Please remove

cGMP, cAMP, Calcium

Manipulation of cGMP, cAMP or Calcium levels is a highly promiscuous and aspecific approach. Their manipulation results in the modification of hundreds of cellular and physiological processes independent from Eph signaling. Moreover, modification of cAMP,cGMP or calcium levels can be induced by hundreds of receptors and proteins including pumps and G-protein-coupled receptors which play the major role in their regulation rather than Eph kinases playing a minor role.

This section MUST BE REMOVED

CONCLUSION

PAg 20 lines 16-21 the sentence” Modulating second messenger signaling  involved in the epha pathway using recently developed tools enabling subcellular-specific control of these signaling molecules offers alternative strategies with an enhanced specificity for ephrin-a signaling than the alteration of the camp, cgmp or calcium signaling spread over the whole cell. further understanding of the second messenger signals controlling epha signaling might pave the way to further specificity for cyclic nucleotide- and calcium-approaches.” This sentence must be removed accordingly with previous comments on cGMP, cAMP, Calcium

Author Response

The review deals with the manipulation of EphA forward signaling through the use of multiple approaches.
Overall, the paper is unbalanced through aspecific and questionable targeting approaches and the chapter dealing with the signaling pathways is uncomplete and misleading. In fact minor pathways of Eph/ephrin signaling are represented while major pathways are missing. This representation leads the reader to think that you can manipulate calcium, cAMP and cGMP levels to finally modulate EphA signaling. Whilst, obviously, calcium , cAMP and cGMP levels are regulated by many physiological systems and are downstream to many effectors mediating a large number of actions. Ephrins play a minor role only in this story.
Accordingly, targeting approaches on cAMP, cGMP and calcium, which are highly aspecific, not selective and promiscuous must be removed.

We thank the reviewer for the time spent evaluating our manuscript. We have deeply remodeled the manuscript to implement the reviewer suggestions. In particular, the cAMP, cGMP and calcium section has been extensively rewritten to highlight the approaches that enable to enhance the specificity for the ephrin-A:EphA pathway when manipulating these signaling molecules.

IN DETAIL:
The title is somehow misleading because forward signaling is triggered by Eph kinase activation whereas reverse signaling is due to ephrins.

We adapted the title to highlight that our manuscript deals with the forward direction of the ephrin-A:EphA signaling pathway

The second paragraph “ EphrinA- forward signalling” is incomplete and does not describe the following:
ï‚· The activation of the receptor through ephrin soluble ligands,
ï‚· The activation via “cis” interaction
ï‚· The activation through heterodimerization with other RTKs (i.e. EGFR or ErbB2)
ï‚· The Kinase dead receptor EphA10
Moreover, even if the Authors support the idea of cAMP,calcium or ephexin targeting, they should describe the more prominent second messengers involved in EPh forward signaling in this section. Figure 1 should be revised accordingly.

We completed the second paragraph following the reviewer suggestions, including the role of soluble ligand, diverse modalities of receptor interactions, the unconventional EphA10 member of the EphA family, and extended the description of the involvement of RhoGTPases downstream of RhoGEFs. This changes are implemented in page 3 and 4, throughout section 2 “Ephrin-A forward signaling pathways”.

PAg 4 lines 4-6 The following sentence “Within the toolset available to interfere with ephrin-A:EphA forward signaling, many strategies target EphA receptors. This abundance might reflect the higher accessibility of EphAs to drugs compared to their intracellular downstream effectors.” is unclear, it must be revised and better explained. It looks like the Authors would suggest a connection between the accessibility of the binding site and the downstream effectors. However, I am not aware of such a kind of correlation.
Moreover, the presence of kinase inhibitors is not surprising since the catalytic domain of kinases is highly conserved and kinase inhibitors are highly promiscuous blocking kinase activity of a large number of kinases. The presence of biological (siRNA, antibodies) is not even surprising since you can project a siRNA for any protein in our body.

The real surprise is the possibility of inhibiting Eph/ephrin interaction with small molecules. In fact protein/protein interaction is usually mediated through the interaction of large surfaces of the receptor and the ligand making the small molecules useless. On the contrary Eph/ephrin interaction needs the insertion of ephrin G-H loop in the binding site of the Eph receptor.

The suggestion of “a connection between the accessibility of the binding site and the downstream effectors” was not in our intent. To clarify the message of this introductive paragraph, we removed the following sentence “This abundance might reflect the higher accessibility of EphAs to drugs compared to their intracellular downstream effectors” (page 5, line 6).

ANTIBODIES:
PF-06647263 and MEDI547 should be included

These antibodies were included in the first version of the manuscript, although under a different name. We clarified this point in the revised manuscript (page 5, line 20 and 27).

iRNAs
This long paragraph is mainly a general overview on iRNAs and I am not sure it is useful to include it in the paper. Probably, I would summarize it by directly explaining their in vivo delivery drawbacks

We agree that the limitations of the different types of iRNAs are not specific to the EphA pathway. They however apply to this pathway as well and we think it is important to clarify the specifics of each approach in this paragraph. This is the reason why we kept this paragraph in the revised manuscript.

Small molecules
Pag 8 line 31: The sentence:” Despite the extensive work done for the last decades, small molecules still appear as a strategy lacking specificity” is correct when talking about kinase inhbitors but it must be corrected when talking of PPI-inhibitors. Even if small molecules PPI-i are unable to discriminate between Eph kinases classes the do not interact with other kinases or target. I suggest to change with ” Despite the extensive work done for the last decades, kinase inhibitors still appear as a strategy lacking specificity

We agree that PPI inhibitors and kinase inhibitors do not achieve the same level of specificity. We clarify this point in page 9, line 11

Ephexin-1
The regulation of ephexins could be an interesting approach in Eph targeting. However, it is unclear why EPhexin-1 only was included in the paper when Ephexin3 has been shown to interact with EphA4, ephexin4 with EphA2 and ephexin5 with EphA4. I suggest to include them in the paper.

We thank the reviewer for her/his suggestion. Ephexin3, 4 and 5 are now included in the manuscript, both in section 2 describing the EphA downstream pathway and in their specific section describing the approaches to manipulate them (page 4, line 10 and page 17, line 20).

Pag 15 line 20: targeting src is far to be a specific approach since src regulates many other proteins. Consequently, the use of the suggested src inhibitors will result in the modulation of many other target other the EPh/ephrin
Pag 15 line 24: PP2 is not a specific src inhibitor since it blocks the activity of many kinase at similar concentrations (PMID: 22594480)
Pag 15 line 27 Since Authors state that “…these approaches are neither specific for Ephexin-1 nor for Src among the SFK family” I suggest to widely revise lines 21-26 in a more consistent way

We agree with the reviewer that manipulating SFKs lacks specificity for the Ephexin pathway and removed this paragraph from the manuscript (page 17, line 28).

Vav2 and vav3
They are downstream of many kinases since the can not be regarded as an approach to modulate ephrin-A forward signaling. Please remove

We thank the reviewer for her/his reminder that Vav2 and 3 involvement is not restricted to the EphA pathway. They are, however, key for this signaling cascade as well and approaches to manipulate them might be crucial to understand how EphA signaling works. Our manuscript does not mention only the EphA effectors that are exclusive to this cascade, and we think that describing strategies to manipulate them might help better understanding how EphAs signal. We agree that these approaches will not be useful to design therapeutic strategies, but they can be key to extend our knowledge of EphA signaling. For instance, identifying the cellular processes requiring each branch of EphA signaling will benefit from manipulation strategies targeting Vav2 and 3. For these reasons, we decided to keep the Vav-focused paragraph.

cGMP, cAMP, Calcium
Manipulation of cGMP, cAMP or Calcium levels is a highly promiscuous and aspecific approach. Their manipulation results in the modification of hundreds of cellular and physiological processes independent from Eph signaling. Moreover, modification of cAMP,cGMP or calcium levels can be induced by hundreds of receptors and proteins including pumps and G-protein-coupled receptors which play the major role in their regulation rather than Eph kinases playing a minor role.
This section MUST BE REMOVED

We respectfully disagree with the reviewer. Like Vav-oriented approaches, manipulating cGMP, cAMP and calcium can help better understand the EphA downstream pathways. We however agree that manipulating these signaling molecules can be highly aspecific. Recently developed strategies enable to enhance the specificity of cGMP, cAMP and calcium manipulation for EphA signaling, by restricting the control of their concentration in subcellular compartments where they are involved in this pathway. We deeply remodeled these paragraphs to focus only on the approaches with enhanced EphA specificity (from page 18, line 28 to the end of page 19).

CONCLUSION
PAg 20 lines 16-21 the sentence” Modulating second messenger signaling involved in the epha pathway using recently developed tools enabling subcellular-specific control of these signaling molecules offers alternative strategies with an enhanced specificity for ephrin-a signaling than the alteration of the camp, cgmp or calcium signaling spread over the whole cell. further understanding of the second messenger signals controlling epha signaling might pave the way to further specificity for cyclic nucleotide- and calcium-approaches.” This sentence must be removed accordingly with previous comments on cGMP, cAMP, Calcium

Accordingly to our previous comments, we kept this sentence intact since it addresses and explains the specificity limits of cGMP, cAMP and calcium manipulations.

You can also see the attachment.

Reviewer 2 Report

The manuscript describes how manipulating ephrin-A:EphA signaling can be achieved by a wide range of approaches and tools. The authors provide a detailed account of established and new methods in this regard. Section 1 is rather cursory and lacks detail when different organs systems are discussed. The section on second messenger systems appears to be out of place because it describes all the things that these messengers do and how their signaling can be manipulated without direct relevance for how to specifically manipulate ephrin-A:EphA signaling.

The table is a great summary of the literature and data discussed in the paper.

Page 1, Line 28: change: ‘EphA4 binding ephrin-B2’ to ‘EphA4 which binds ephrin-B2’

P2, L20: Elaborate on apoptosis and caspase pathways.

P2, L33: Section 1.3 is interesting, but the individual organs or tissues are handled in a very brief manner. Maybe you can elaborate, e.g., kidney, endocrine pancreas (insulin), etc.

P3, L2: ‘Several members of the ephrin-A system including EphA1, EphA4, ephrin-A1 and ephrin-A5 have been implicated in a diversity of neurodegenerative conditions such as Alzheimer disease or amyotrophic lateral sclerosis’ How so, please elaborate?

P5, L3: change ‘exhibit’ to ‘exhibits’

P5, L6: explain ‘camelids’

P5, L41: change ‘underlie’ to ‘underlies’

P17, L37: here you mention EphB. Other than the first sentence of the Introduction, the reader does not know anything about EphB.  Can you elaborate on EphB?

P19, L1: change ‘signals’ to ‘signal’

P20, L21: change ‘as well as a the’ to ‘as well as the’

P18, Section 5: The section of ‘Targeting second messengers’ and the discussion of cAMP, cGMP and calcium makes it clear that these are general second messengers.  Even though, they are involved in downstream signaling effects of EphA, a direct functional link to EphA is difficult to use in a clinical scenario. Therefore, this section is not necessary to explain their general function in a review about ephrin-A:EphA forward signaling. The authors themselves point out these limitations in the section.

P22, L4: change ‘range of approach’ to ‘range of approaches’

Author Response

The manuscript describes how manipulating ephrin-A:EphA signaling can be achieved by a wide range of approaches and tools. The authors provide a detailed account of established and new methods in this regard. Section 1 is rather cursory and lacks detail when different organs systems are discussed. The section on second messenger systems appears to be out of place because it describes all the things that these messengers do and how their signaling can be manipulated without direct relevance for how to specifically manipulate ephrin-A:EphA signaling.
The table is a great summary of the literature and data discussed in the paper.

We thank the reviewer for her/his kind assessment of our manuscript. Following the reviewer suggestion we detailed further section 1 (see details below) and limited the extent of the second messenger section. We clarified how this generic signaling molecules can be manipulated with enhanced specificity for ephrin-A:EphA signaling.

P2, L20: Elaborate on apoptosis and caspase pathways.

The interaction between the ephrin-A pathway an apoptosis has been clarified, highlighting the control of the neuronal progenitor pool during development (page 2, line 22).

P2, L33: Section 1.3 is interesting, but the individual organs or tissues are handled in a very brief manner. Maybe you can elaborate, e.g., kidney, endocrine pancreas (insulin), etc.

The paragraph describing the role of ephrin-A forward signaling outside the nervous system has been extended to clarify its involvement in the different organs mentioned (page 2).

P3, L2: ‘Several members of the ephrin-A system including EphA1, EphA4, ephrin-A1 and ephrin-A5 have been implicated in a diversity of neurodegenerative conditions such as Alzheimer disease or amyotrophic lateral sclerosis’ How so, please elaborate?

The impact of EphAs on the Alzeimer’s disease and amyotrophic lateral sclerosis is explained further on page 3, line 14.

P5, L6: explain ‘camelids’

The reason why this family of animals that includes camels and llamas are interesting for the generation of nanobodies is now explained on page 6, line 5.

P17, L37: here you mention EphB. Other than the first sentence of the Introduction, the reader does not know anything about EphB. Can you elaborate on EphB?

EphBs are mentioned is the introductory paragraph as one of the two classes of Eph receptors. Since the focus of this review is the ephrin-A:EphA pathway, we think that describing further EphB might rather confuse the reader.

P18, Section 5: The section of ‘Targeting second messengers’ and the discussion of cAMP, cGMP and calcium makes it clear that these are general second messengers. Even though, they are involved in downstream signaling effects of EphA, a direct functional link to EphA is difficult to use in a clinical scenario. Therefore, this section is not necessary to explain their general function in a review about ephrin-A:EphA forward signaling. The authors themselves point out these limitations in the section.

We acknowledge that explaining the general function of second messengers is not necessary in this review. We deeply rewrote the second messenger section that now focuses on the interaction of these signaling molecules with the ephrin-A pathway (from page 18, line 28 to the end of page 19). We acknowledge that this enhanced specificity might not be sufficient for clinical use. However, these tools can be useful for a better understanding of the EphA dowstream events.

Page 1, Line 28: change: ‘EphA4 binding ephrin-B2’ to ‘EphA4 which binds ephrin-B2’
P5, L3: change ‘exhibit’ to ‘exhibits’
P5, L41: change ‘underlie’ to ‘underlies’
P19, L1: change ‘signals’ to ‘signal’
P20, L21: change ‘as well as a the’ to ‘as well as the’
P22, L4: change ‘range of approach’ to ‘range of approaches’

We apologize for these typos and thank the reviewer for her/his careful check. The requested changes have been implemented in the revised manuscript.

You can also see the attachment.

Round 2

Reviewer 1 Report

Thank for thoroughly revising the chapter dealing with the signaling pathways however the figure report minor pathways of Eph/ephrin signaling while major pathways are missing. Please correct.

Table 1 page 10:

About peptides: the sentence“Tissue penetration compatible with in vivo experimentation” is inconsistent with “Low stability and Low pharmacokinetics and bioavailability in vivo”. You probably mean that peptides are compatible with in vitro experimentation

About PPI-inhibitors the sentence “Weak stability and bioavailability in Vivo”is not correct. It depends on the small molecule. I.e. UniPR1331, UniPR139 and UniPR500 are stable and orally bioavailable

About PPI-I vs kinase inhibitors: both of them are suggested as “low specific”. However, Kinase inhibitors are unable to discriminate within kinases in general whereas PPI-I are unable to discriminate within Eph-kinases. Please correct

About cAMP, cGMP and calcium: “low specificity” must be added within limits, since they are not more specific than kinase or PPI-inhibitors

cAMP, cGMP and calcium

Even if the Authors claim to an improved specificity through lipid raft targeting it remains a highly aspecific approach. In fact the role, function and existence of lipid raft it is still largely debated (Trends Cell Biol. 2020 May;30(5):341-353.). Moreover, lipid rafts are not exclusive for Eph/ephrins, in fact many proteins have been identified in lipid raft.

Spongee is a brilliant approach to selectively modulate cGMP levels without affecting cAMP concentration (ref 78) but no data on Eph system has been reported in ref 78.

Similarly for Calcium and ref 80.

Finally, as reported by the Author in ref 78 and : “SponGee (Spicee) enables the investigation of local cGMP signals in vivo and in discrete subcellular domain”. Modifying cGMP signaling in discrete subcellular domain does not mean specifically targeting Eph system, such a targeting still involve dozens of receptors/enzymes.

 In my opinion the paper should be acceptable only after removing the section on cGMP, cAMP and calcium.

Author Response

Thank for thoroughly revising the chapter dealing with the signaling pathways however the figure report minor pathways of Eph/ephrin signaling while major pathways are missing. Please correct.

The Akt and E-Cadherin pathways have been added to Figure 1, as suggested by the reviewer in an email clarifying her/his request.

Table 1 page 10:
About peptides: the sentence“Tissue penetration compatible with in vivo experimentation” is inconsistent with “Low stability and Low pharmacokinetics and bioavailability in vivo”. You probably mean that peptides are compatible with in vitro experimentation
About PPI-inhibitors the sentence “Weak stability and bioavailability in Vivo”is not correct. It depends on the small molecule. I.e. UniPR1331, UniPR139 and UniPR500 are stable and orally bioavailable
About PPI-I vs kinase inhibitors: both of them are suggested as “low specific”. However, Kinase inhibitors are unable to discriminate within kinases in general whereas PPI-I are unable to discriminate within Eph-kinases. Please correct
About cAMP, cGMP and calcium: “low specificity” must be added within limits, since they are not more specific than kinase or PPI-inhibitors

We thank the reviewer for these comments and adjusted Table 1 following the reviewer suggestions:
- Peptides: we restrict our claim to the efficient tissue penetration of these tools
- PPI-inhibitors: “Weak stability and bioavailability in Vivo” has been removed from the table
- PPI-I vs kinase inhibitors: we clarified the level of specificity for both these approaches
- cAMP, cGMP and calcium: low specificity has been added within limits

cAMP, cGMP and calcium
Even if the Authors claim to an improved specificity through lipid raft targeting it remains a highly aspecific approach. In fact the role, function and existence of lipid raft it is still largely debated (Trends Cell Biol. 2020 May;30(5):341-353.). Moreover, lipid rafts are not exclusive for Eph/ephrins, in fact many proteins have been identified in lipid raft.

We are aware of the controversy about lipid rafts. However, this controversy is now more focused on the convergence between the membrane domains purified using detergent-based biochemical methods (now called DRM for detergent resistant membrane), rather than on the existence of membrane domains within the plasma membrane of living cells. For instance, the reference mentioned by the reviewer to support the idea that "the role, function and existence of lipid raft it is still largely debated" starts its concluding remarks by "The studies described in the earlier section have provided compelling evidence that strongly supports the existence of rafts in vivo". Still, we agree that visualizing rafts would be a great advance to close this controversy, but it is far from the focus of our manuscript.

Spongee is a brilliant approach to selectively modulate cGMP levels without affecting cAMP concentration (ref 78) but no data on Eph system has been reported in ref 78.
Similarly for Calcium and ref 80.

We respectfully disagree with the reviewer assertions. Figure 6F of reference 78 describes a set of data demonstrating that SponGee prevents ephrin-A5-induced axon retraction when targeted to lipid rafts but not when it is excluded from this plasma membrane compartment. This observation is further supported by Figure S6 of the same article. In addition, Figure S5 describes an elevation of cGMP when axons are exposed to ephrin-A5, and the blockade of this elevation when SponGee is expressed.
Similarly, Figure 6F of reference 80 describes a set of data demonstrating that SpiCee prevents ephrin-A5-induced axon retraction when targeted to lipid rafts but not when it is excluded from this plasma membrane compartment. These observations are further supported by Figures S5 and S6 of the same reference.

Finally, as reported by the Author in ref 78 and : “SponGee (Spicee) enables the investigation of local cGMP signals in vivo and in discrete subcellular domain”. Modifying cGMP signaling in discrete subcellular domain does not mean specifically targeting Eph system, such a targeting still involve dozens of receptors/enzymes.

We agree with the reviewer that restricting the manipulation of these signaling molecules to lipid rafts does not provide absolute specificity and this is the reason why we only claim that the specificity is only enhanced compared to general manipulation of these second messengers. For instance we state in Page 15, Line 38, when referring to lipid raft-specific manipulations of cGMP, cAMP and Ca2+: "Controlling second messenger concentration in this cellular domain provides approaches to manipulate the ephrin-A:EphA pathway with enhanced specificity compared to altering cAMP, cGMP and calcium in the entire cell." This is also the reason why we precise in our concluding remarks that "Further understanding of the second messenger signals controlling EphA signaling might pave the way to further specificity for cyclic nucleotide- and calcium-approaches" (Page 20, Line 10).
The same limited specificity stands for other approaches like the kinase inhibitors for instance, a manipulating approach described in our manuscript and that is deemed acceptable by the reviewer.
We further clarified the limited specificity of second messenger-based approaches in table 1, as suggested by the reviewer.

In my opinion the paper should be acceptable only after removing the section on cGMP, cAMP and calcium.

We respectfully disagree with the referee and think that these approaches can be part of this review since they enable to interfere with ephrin-A:EphA signaling.

You can also see the attachment.
